# Broad Antifungal Spectrum of the Pore-Forming Peptide C14R Against *Cryptococcus* and *Candida* Species from the WHO Fungal Priority Pathogens List

**DOI:** 10.3390/pathogens14060511

**Published:** 2025-05-22

**Authors:** Carolina Firacative, Norida Vélez, Ann-Kathrin Kissmann, Daniel Alpízar-Pedraza, Jan-Christoph Walter, Ludger Ständker, Frank Rosenau

**Affiliations:** 1Studies in Translational Microbiology and Emerging Diseases (MICROS) Research Group, School of Medicine and Health Sciences, Universidad de Rosario, Bogota 111221, Colombia; noridavelezc@gmail.com; 2Institute of Pharmaceutical Biotechnology, Ulm University, 89081 Ulm, Germany; ann-kathrin.kissmann@uni-ulm.de (A.-K.K.); daniel.alpizar@cidem.cu (D.A.-P.); jan-christoph.walter@uni-ulm.de (J.-C.W.); frank.rosenau@uni-ulm.de (F.R.); 3Center for Pharmaceutical Research and Development (CIDEM), 26th Avenue, No. 1605, Nuevo Vedado, La Habana 10400, Cuba; 4Core Facility for Functional Peptidomics (CFP), Faculty of Medicine, Ulm University, 89081 Ulm, Germany; ludger.staendker@uni-ulm.de

**Keywords:** *Candida*, cryptococcosis, *Cryptococcus*, invasive candidiasis, antimicrobial peptides, C14R, antifungal resistance

## Abstract

The World Health Organization (WHO) prioritized 19 fungal species based on the significant impact of these pathogens on human health, including the emergence of antifungal resistance, which highlights the necessity of finding new antifungal therapies. Among these novel therapeutic approaches, the antimicrobial pore-forming peptide C14R has shown to be promising against *Candida albicans* and *Candida auris*. In this study, the antifungal in vitro efficacy of C14R was assessed against six additional species from the WHO priority list, *Cryptococcus neoformans*, *Cryptococcus gattii*, *Candida glabrata*, *Candida tropicalis*, *Candida parapsilosis* and *Candida krusei*, as well as against *Candida dubliniensis.* This study shows that C14R has good antifungal activity against several clinical isolates of the studied species, with MIC values between 0.8476 and 10.88 µg/mL. Most notably, some of the studied isolates are resistant to commonly used antifungal drugs but are susceptible to the peptide. C14R showed, moreover, its capacity to disrupt *Cryptococcus* capsules, beyond its already proven capacity to disrupt plasma membranes, and its antifungal activity was not affected depending on the serotype or species assessed. The inclusion of basidiomycete and ascomycete yeasts allowed us to display the broad-spectrum potential of C14R, highlighting it as a promising candidate as an antifungal agent.

## 1. Introduction

In December 2022, the World Health Organization (WHO) released the first priority list of fungal pathogens that can cause invasive acute and subacute systemic mycoses. This list, including 19 species, was built based on the significant impact of these pathogens on human health, considering mainly disease burden-related criteria, such as mortality, annual incidence and morbidity, as well as the emergence of antifungal resistance and the major knowledge gaps on the global burden of fungal infections [1].

From the prioritized species, there are only two among the basidiomycete yeasts, *Cryptococcus neoformans* and *Cryptococcus gattii*. It is of notice *C. neoformans* is listed first in the ‘critical threat’ group, which includes those pathogens ranked highest for perceived public health importance, while its sibling species, *C. gattii*, is positioned in the ‘medium threat’ group. Unique to these human pathogens is the presence of a polysaccharide capsule, which not only contributes to environmental survival but is also a key virulence factor for mammalian pathogenesis [2]. Widely distributed in the environment, these encapsulated yeasts are the etiological agents of cryptococcosis, the third most frequent invasive mycoses in the world, presenting mainly as meningoencephalitis, with high mortality rates despite antifungal therapy [3]. Prevailing among the HIV/AIDS population, it is estimated that every year, 152.000 cases of cryptococcal meningitis occur in the world, with 112.000 deaths, which represents about 19% of AIDS-related mortality [4]. Furthermore, the frequency of cryptococcosis among HIV negative population, including solid organ transplant recipients and others receiving exogenous immunosuppression, is continuously increasing, which brings unique diagnostic and therapeutic challenges, since there are no clear guidelines for non-HIV at-risk groups [5]. Another great concern regarding cryptococcosis management are the limited therapeutic options resulting from the toxicity of amphotericin B deoxycholate, a first-line antifungal, the reduced susceptibility and heteroresistance to fluconazole which have long been described in both cryptococcal species, and the restricted availability of pricey antifungals such as liposomal amphotericin B and 5-fluorocytosine [6,7]. In addition, other antifungals like the echinocandins are ineffective in the treatment of *Cryptococcus* infections due to their intrinsic resistance to these drugs [8].

Among the ascomycete yeasts in the fungal priority list, *Candida glabrata* (re-classified as *Nakaseomyces glabratus*), *Candida tropicalis* and *Candida parapsilosis* are listed in the ‘high threat’ group, while *Candida krusei* (re-classified as *Pichia kudriavzevii*) is in the ‘medium threat’ group. Together, these globally distributed pathogenic yeasts are responsible for a significant proportion of cases of invasive candidiasis worldwide, with an ongoing increase in their prevalence [9,10,11,12]. Invasive candidiasis, comprising both candidemia and deep-seated infections, is commonly associated with health care settings and one of the most frequent invasive mycoses in the world, with mortality rates that many times exceed 40% [13]. Although *Candida albicans* remains the most common etiological agent of invasive candidiasis, the growing incidence of other species is a public health concern, since many non-*albicans Candida* species are able to rapidly acquire antifungal resistance or are intrinsically resistant to azoles and to a lesser extent to echinocandins [14]. Moreover, the mortality rates of infections caused by non-*albicans Candida* species could be as high as 64%, similar to that reported in cases of *C. glabrata* candidemia [1,15,16,17]. Adding to the aforementioned is the specific pathogenicity feature that many *Candida* species have, which is the ability to form biofilms that protect them from external factors such as the host’s immune system and antifungal drugs [18].

Although the treatments for cryptococcosis and invasive candidiasis are generally well-established, some antifungals are still unavailable in many countries [19,20]. Together with toxicity, insufficient bioavailability of some medications that are used orally and the emergence of antifungal resistance [21,22], there is therefore a clear need to search for alternative treatments to fight systemic infections caused by both *Cryptococcus* and *Candida* species. Either as standalone or in combination therapies, antimicrobial peptides (AMPs), which are small cationic peptides that are part of the innate immune systems of different organisms, hold great promise in the treatment of life-threatening mycoses [23,24]. Among these molecules, C14R, which is a designed analog derived from the peptide BP100 isolated from bee venom, was recently reported to have a potent antifungal activity against a large set of clinical isolates of the prevalent species *C. albicans* and the emergent multidrug-resistant *Candida auris* (re-classified as *Candidozyma auris*), including fluconazole and amphotericin B-resistant isolates [25]. In addition, C14R had a synergistic effect when combined with fluconazole, was able to inhibit the biofilm and growth of both *Candida* species and had the capacity to disrupt *Candida* membranes [25]. Notably, *C. albicans* and *C. auris* are also part of the ‘critical threat’ group of the WHO fungal priority pathogens list [1]. Further, C14R was able to negatively affect the viability of *C. parapsilosis* by pore formation, as well as to halt biofilm maturation and significantly reduce the biomass of preformed biofilms by over 70% [26]. The limited toxicity against human cells, the anti-inflammatory properties and the ability to modulate immune responses are characteristics of C14R that add to its advantages as a candidate to treat fungal infections [27,28].

In this study, the extended in vitro antifungal activity of C14R is demonstrated, not only against the basidiomycete encapsulated yeasts *C. neoformans* and *C. gattii* but also against additional *Candida* species, including *C. glabrata*, *C. tropicalis, C. parapsilosis*, *C. krusei*, and the less commonly recovered *Candida dubliniensis*. It is also shown that apart from the capacity of C14R to form pores in yeast plasma membranes, as previously reported, the peptide is able to cross the complex capsule of *C. neoformans*. Additionally, C14R is shown to strongly interact with the polysaccharide capsule of both cryptococcal species, independent of the serotypes A or C. Remarkably, most of the studied isolates were recovered from cases of invasive infection, such as cryptococcal meningitis and candidemia, and include fluconazole-resistant isolates. The inclusion of ten or more isolates of the species *C. glabrata*, *C. tropicalis*, *C. parapsilosis*, *C. neoformans* and *C. gattii* strengthen this study. In conclusion, and considering previous findings from our group [25,26], the current study shows that the peptide C14R is a promising therapeutic option for further development as an antifungal agent to treat infections caused by eight of the 19 WHO-prioritized fungal pathogens.

## 2. Materials and Methods

### 2.1. Isolates

Twenty isolates of *Cryptococcus* and sixty-eight of *Candida* were included in this study. From the isolates, 28 (31.8%) were *C. parapsilosis*, 18 (20.4%) were *C. glabrata*, 17 (19.3%) were *C. tropicalis*, 10 each (11.4%) were *C. neoformans* (serotypes A and D) and *C. gattii* (serotypes B and C), four (4.5%) were *C. dubliniensis* and one isolate (1.1%) was *C. krusei*. Most isolates (97.7%) belong to the culture collection of the MICROS Research Group of Universidad del Rosario and are stored at −80 °C in 10% glycerol. One isolate of *C. parapsilosis,* ATCC^®^ 22019, and the only isolate of *C. krusei,* ATCC^®^ 6258, are reference strains. Data on the minimum inhibitory concentration (MIC) to 5-fluorocytosine, posaconazole, voriconazole, itraconazole, fluconazole and amphotericin B, were available for all the studied isolates [17,29,30,31]. *Candida* isolates had additional data on the antifungal susceptibility to the echinocandins anidulafungin, micafungin and caspofungin [17]. Susceptible or resistant isolates were established according to the Clinical and Laboratory Standards Institute (CLSI) breakpoints and epidemiological cutoff values per species [32,33,34]. From the isolates, 16 (18.2%) were resistant to fluconazole, including four each of *C. parapsilosis* and *C. gattii*, three of *C. neoformans*, two each of *C. glabrata* and *C. tropicalis* and the isolate of *C. krusei*. From these, one of the isolates of *C. tropicalis* and one of *C. neoformans* were concomitantly resistant to voriconazole. One *C. glabrata* isolate was resistant to caspofungin, and three and one of *C. tropicalis* and *C. krusei*, respectively, were resistant to 5-fluorocytosine (Appendix A).

Prior to the evaluation of the antifungal susceptibility testing, all isolates were cultured on Sabouraud dextrose agar and incubated for 24 to 48 h at 35 °C.

### 2.2. Peptide

C14R was synthetized commercially by Synpeptide Co., Ltd. (Shanghai, China) with a purity of 95%. The peptide, consisting of 16 amino acid residues, CSSGSLWRLIRRFLRR, with a molecular weight of 2006.37 g/mol [28], was received in a lyophilized form. At arrival, a stock solution of the peptide with a concentration of 400 µg/mL was prepared in 4 mL of sterile water and frozen at −20 °C. Immediately before the experiments, the stock solutions were thawed.

### 2.3. Antifungal Susceptibility Testing

Using broth microdilution, the antifungal activity of C14R against *Cryptococcus* and *Candida* isolates was determined as previously reported [25], following the CLSI guidelines, M27M44S protocol [32]. The concentration of the peptide ranged between 0.390625 to 200 µg/mL, tested by 2-fold serial dilutions. All microdilution plates, containing a yeast inoculum of 1–5 × 10^3^ cells/mL plus the different concentrations of the peptide, in a final volume of 200 µL per well, were incubated at 35 °C. The MIC of C14R, defined as the lowest concentration of the peptide that significantly decreased the yeast growth (>50%) compared to the growth control, was determined visually. Plates with *Cryptococcus* isolates were read after 72 h of incubation and those with *Candida* isolates after 24 h.

### 2.4. In Silico Study of C14R-Capsule Interaction

For the interaction study between C14R and the *Cryptococcus* capsules, serotypes A and C were selected to represent *C. neoformans* and *C. gattii*, respectively. The capsules were constructed following the specifications previously reported, with minimal modifications [35]. Both serotypes share a main chain composed of (1 → 3) glycosidic-linked α-D-mannopyranosyl (α-D-Man*p*) residues. A triad unit (MC1−MC2−MC3) consists of three α-D-Man*p* residues. Regardless of the serotype, a β-D-glucopyranosyluronic acid (β-D-GlcA*p*) sidechain residue (SC1) is linked to MC1 via a (1 → 2) glycosidic bond. In serotype A, two β-D-xylopyranosyl (β-D-Xyl*p*) residues (SC2 and SC3) are attached to MC2 and MC3, respectively, through a (1 → 2) glycosidic bond. In serotype C, in addition to these β-D-Xyl*p* residues, two extra β-D-Xyl*p* residues (SC4 and SC5) are linked to MC1 and MC3 through a (1 → 4) glycosidic bond [36] (Figure 1). For each serotype, the mannose triad was repeated 18 times, with an α-D-Man*p* cap to the first triad and a terminal hydroxyl group at the reducing end. Additionally, the O6 atoms of MC1 and MC3 were acetylated [37,38]. The oligosaccharides were constructed using the Glycan Reader & Modeler tool integrated into the CHARMM-GUI web server [39,40,41,42]. The interaction study was conducted using a single monomer with 18 repeats and a scaffold of 25 monomers in a 5 × 5 array to simulate a more complex capsule for each serotype. The distances between polysaccharides were set to 1.5 nm, similar to other reports [35]. The 3D structure of C14R was obtained from previous studies [28,43]. The C14R-capsule complexes were predicted using a Monte-Carlo (MC) iterated search combined with the BFGS gradient-based optimizer implemented in AutoDock Vina [44,45]. The conformational search space was specified by a cubic box previously defined using AutoDock Tools, each of them surrounding the serotype system in study. The docking simulations were conducted considering all rotatable angles of the C14R sidechain residues as flexible and carbohydrates as rigid. All docking parameters were set to default (hydrophobic_dist_cutoff = 4.0, hydrogen_bond_angle_cutoff = 40.0, hydrogen_bond_dist_cutoff = 4.0, salt_bridge_dist_cutoff = 5.5) except for exhaustiveness, which was increased to 50 to ensure a thorough search. For the in silico docking experiments, the Python-implemented computer algorithm BINANA, version 2.7 (BINding ANAlyzer, (National Biomedical Computation Resource, La Jolla, CA, USA)) was used [46].

### 2.5. C14R Permeabilization Assay

The pore formation capability of C14R was already demonstrated in *C. albicans* membrane [25], which is comparable to the membrane of other *Candida* species [47]. However, in this study, an additional permeabilization assay was performed to demonstrate the ability of the peptide to cross not only the plasma membrane but also the polysaccharide capsule of *C. neoformans*, a unique characteristic of species of the genus *Cryptococcus* [36,48]. For this, four fluorescent dyes with molecular sizes ranging from 389 to 1250 Da (fluorescein (FITC) < propidium iodide < ATTO 488 alkyne < rhodamine phalloidin) (Thermo Fisher Scientific Inc., Schwerte, Germany) were used at a final concentration of 5 µL/mL as described before [25,28]. In brief, 10^7^ cells of *C. neoformans* (DSM 11959) were incubated for 2 h at 37 °C in 200 µL RPMI-1640 media supplemented with 25 µg/mL of C14R, which is equivalent to one dilution more than the MIC value. After incubation, the cells were washed with phosphate-buffered saline (PBS) and treated with 5 µL of each fluorescent dye, for 20 min. A solution of paraformaldehyde was used for 10 min in order to fixate the cells [25,28]. Cells remained untreated for negative controls and cells treated with the pore-forming detergent Triton X-100 (0.4%) served as positive controls. The fluorescence of treated and untreated cells was measured at excitation wavelengths of 498 nm (FITC), 535 nm (propidium iodide), 500 nm (ATTO 488 alkyne) and 540 nm (rhodamine phalloidin), and emissions of 517 nm (FITC), 617 nm (propidium iodide), 520 nm (ATTO 488 alkyne) and 565 nm (rhodamine phalloidin) with a Tecan SPARK microplate reader (Tecan Group Ltd., Männedorf, Switzerland). Fluorescence values were corrected and normalized to the background signal obtained by C14R-untreated cells. Experiments were conducted as triplicate.

### 2.6. Statistical Analysis

The frequency of MICs, mode and geometric mean MICs of C14R were determined per each species of *Cryptococcus* and *Candida*. The differences in MICs between species with ten or more isolates were established using a Mann–Whitney test. The Pearson correlation coefficient (ρ) was used to assess the association between the MIC values of the peptide and those of fluconazole, considering the inclusion of 16 isolates that are resistant to this azole. Given that most of the studied isolates were susceptible to the other antifungals, no other test for association between MIC values was carried out. In the permeabilization assay, differences in fluorescence between the treated and untreated cells were determined using Student’s t-test. Statistical analyses were performed with GraphPad Prism 9.4.1 (La Jolla, CA, USA). When calculated, *p*-values <0.05 were considered statistically significant.

## 3. Results

### 3.1. C14R Has a Good in Vitro Antifungal Effect Against Cryptococcus and Candida Isolates

Generally, C14R had a good antifungal activity against the seven evaluated species. The geometric mean minimum inhibitory concentration (MIC) of the peptide ranged between 0.8476 and 10.88 µg/mL, with modes ranging between 0.390625 µg/mL and 12.5 µg/mL (Table 1). Notably, none of the isolates of *Cryptococcus* or of *Candida* was able to grow in the two highest concentrations of C14R tested (100 µg/mL and 200 µg/mL).

Assessing each genus separately, it was possible to determine that C14R has, to some extent, an inferior antifungal activity against *Cryptococcus* isolates compared to that observed against *Candida* isolates. While the mode of C14R against both cryptococcal species was 12.5 µg/mL, the sixth lowest concentration tested, the mode of C14R against all *Candida* species was 0.390625 µg/mL, the lowest concentration tested. In fact, the susceptibility of the isolates of both *C. neoformans* and *C. gattii* to the peptide was statistically lower when compared with each *Candida* species (*p*-values ranging between 0.0054 and <0.0001). On the other hand, even though the geometric mean MIC differed slightly among *Candida* species, no statistical significance was found. Altogether, our results indicate that C14R has a good antifungal activity against *C. neoformans* and *C. gattii* isolates and a potent activity against isolates of *C. parapsilosis*, *C. glabrata*, *C. tropicalis*, *C. dubliniensis* and *C. krusei*.

No association was found between the susceptibility of each evaluated species to C14R and fluconazole (*p* > 0.05). In addition, from all the isolates resistant to commonly used antifungals, only one of *C. glabrata*, which is resistant to fluconazole (MIC of 64 µg/mL), was able to grow at a high concentration of C14R (50 µg/mL). This indicates that the peptide has a good antifungal activity against resistant isolates, as low concentrations of C14R inhibit their growth.

### 3.2. C14R Strongly Interacts with the Cryptococcal Capsule

Previous studies from our group revealed that there are strong interactions between the *Candida* plasma membrane and C14R, when a lipid composition mimicking this membrane was simulated in silico [25]. In the present study, an additional approach was used to assess the interactions of the peptide with the polysaccharide capsule that *C. neoformans* and *C. gattii* have outside their plasma membranes and cell walls. Considering that cryptococcal strains of different serotypes express a range of motifs of the major polysaccharide glucuronoxylomannan (GXM), which makes up 90% of the capsule, the polysaccharide composition mimicking the capsule of two serotypes, A (*C. neoformans*) and C (*C. gattii*), was simulated and served to assess *Cryptococcus*–C14R interactions (Figure 1). A docking approach was therefore employed to determine whether C14R exhibits higher affinities for any of the *Cryptococcus* capsules. In Table 2, the binding energies obtained using the AutoDock Vina scoring function for C14R interacting with both monomers and complexes of each serotype are shown. In general, C14R appeared to have a higher affinity for the carbohydrates forming the capsule of *C. gattii* (−11.526 kcal/mol) compared to those of *C. neoformans* (−9.368 kcal/mol) when analyzed in a monomeric state. This difference may be attributed to the higher number of β-D-xylopyranosyl (β-D-Xyl*p*) branches in the serotype C, which increases the accessibility and arrangement of hydroxyl groups, thereby enhancing hydrogen bonding and other molecular interactions. However, when the peptide interacts with the capsules in a more realistic structural arrangement (complexes), the binding energies show no significant differences (Table 2). This could be because the absence of β-D-Xyl*p* in a single serotype A monomer is compensated by neighboring molecules within the scaffold, allowing similar interactions to occur, as in serotype C.

To identify which residues of C14R play a key role in stabilizing its interaction with *Cryptococcus* capsules, an additional molecular analysis was conducted. In all of the studied complexes, arginine residues were crucial for interactions. C14R is a cationic peptide with five arginine residues while the *Cryptococcus* capsules contain numerous polar hydroxyl groups, facilitating hydrogen bond formation. Furthermore, the presence of β-D-glucopyranosyluronic acid (β-D-GlcA*p*) side chains imparts a negative charge to the capsule, enabling the formation of salt bridges with the positively charged arginine residues of C14R. However, some differences can be observed in the interaction patterns of C14R with each serotype. For serotype A, C14R primarily interacts through residues in the C-terminus region, starting from Arg8. In contrast, with serotype C, the interactions are more evenly distributed across the entire structure of C14R, with Ser2 contributing additionally to the arginines by stabilizing the complexes through polar interactions (Figure 2). These theoretical results suggest that C14R exhibits a similar affinity for interacting with the capsules of both *C. neoformans* and *C. gattii*, highlighting that the interaction between C14R and the capsule is independent of the serotype. Our results can even be extrapolated to the serotype D of *C. neoformans* and B of *C. gattii*, which were not analyzed, considering the composition of their capsules. Serotype D strains have only one β-D-Xyl*p* residue (SC3) attached to MC3, compared to the serotype A strains, which have two residues. Similarly, serotype B strains also have one fewer β-D-Xyl*p* residue (SC5) attached to MC3 than serotype C, which has four residues [36].

### 3.3. C14R Forms Small Pores in C. Neoformans

The pore-forming ability of C14R was initially demonstrated for the human pathogenic Gram negative encapsulated bacterium *Pseudomonas aeruginosa* [28]. In the yeasts *C. albicans* and *C. auris*, particularly, C14R has also shown to cause plasma membrane disruptions due to its capacity to form pores in these biological membranes [25]. This effect was also recently described in the membranes of *C. parapsilosis* [26]. To provide further evidence and validate the predicted pore-forming activity of C14R in other fungi that are less closely related to *Candida* and that in addition present a polysaccharide capsule, a permeabilization assay was conducted using living cells of *C. neoformans*. These cells were exposed to C14R at a concentration of 25 µg/mL in the presence of four fluorescent dyes of varying molecular sizes (rhodamine phalloidin > ATTO488 alkyne > propidium iodide > FITC) alongside untreated cells, which served as a negative control for pore formation and Triton X-100 treated cells serving as positive control (Figure 3).

While in *C. albicans*, dyes up to 741 Da were able to penetrate to the cytoplasm [25], only the smallest dye FITC (389 Da) was able to freely enter the cells of *C. neoformans*. Propidium iodide (668 Da), ATTO488 alkyne (741 Da) and rhodamine phalloidin (1250 Da) remained completely excluded from cell entry with no significant differences observed between treated and untreated cells. This result indicates that C14R not only forms pores in general but also allows us to suggest that there is a defined size restriction for molecules to pass through the formed pores into the cytoplasm of different yeast species.

## 4. Discussion

In view of their global prevalence, the high risk of mortality from the systemic infections they cause, and the threat of antifungal resistance, *C. neoformans*, *C. gattii*, *C. glabrata*, *C. tropicalis*, *C. parapsilosis* and *C. krusei* are among the 19 pathogens of the fungal priority list recently issued by the WHO [1]. Therefore, studies like ours are important to illustrate the potential use of diverse molecules such as AMPs as standalone treatments or in combination with existing antifungals, supporting research into new innovative antifungal therapies which aim to treat resistant species, to prevent further resistance or even to develop more accessible drugs. This study provides evidence on the notable in vitro antifungal activity of C14R, a pore-forming peptide, against both *Cryptococcus* and *Candida* species, suggesting a broad-spectrum efficacy, since these genera belong to two very divergent groups, the basidiomycetes and ascomycetes yeasts, respectively.

*C. neoformans* and *C. gattii* virulence is mediated predominantly by the polysaccharide capsule that surrounds its cell wall, given that this structure has multiple effects on the host’s immune system. The capsule and polysaccharide shedding provide a physical barrier that interferes with normal phagocytosis, phagocytic cell oxidative bursts and clearance by immune cells [2,48]. As such, the use of molecules that may target the capsule, resulting in its inactivation and potentially reduced virulence of these yeasts, can be thought as a good therapeutic strategy. While in many encapsulated microorganisms such as *Klebsiella pneumoniae* and *Streptococcus pneumoniae* the thick coating of polysaccharides confers resistance to AMPs, the level of protection of these capsules can vary, even being null, depending on the specific structure and composition of the capsule itself, as well as on the type of AMP involved [49,50,51]. For instance, the very well-known AMPs polymyxin B and E (also known as colistin) commonly used in the clinic are more effective against encapsulated bacteria because the peptide binds to the lipopolysaccharides and phospholipids in their outer membranes, leading to cell disruption and lysis [51]. In this study, the low MIC values of C14R needed to inhibit the total growth of both *C. neoformans* and *C. gattii* demonstrated the good anti-cryptococcal activity of the peptide. In addition, C14R, which is able to cause membrane disruptions in the encapsulated bacterium *P. aeruginosa* and various species of *Candida*, showed that it has the ability to interact and penetrate a model capsule of typical cryptococcal cells, as indicated by molecular dynamics simulations. Moreover, these theoretical findings were complemented by a critical property of C14R, which is the capacity to form peptide aggregates as amphipathic pore-like structures in the target’s surface, as demonstrated via a permeabilization assay. Although only the smallest fluorescent dye in a series of dyes of increasing molecular weights (ranging from 389 to 1250 Da) was taken up by *C. neoformans* after exposure to C14R, this suggests that the C14R-induced pores have a molecular size cutoff within this range. It remains unclear, however, whether this cutoff is solely size-dependent or if the chemical nature of the dyes could also influence their uptake.

Among the diverse challenges for the development of antifungal drugs against *Cryptococcus* are those posed by the differences in the capsular composition (serotype), taking into account that a particular drug must be effective against all strains [2]. The capsule is not only a very complex and dynamic structure, but each serotype has predominant motifs [36,52]. Our findings show, however, that differences between species and serotypes did not appear to affect the in vitro antifungal activity of C14R, not only because the MIC values of the peptide were similar in the isolates of both species and the four serotypes (A, B, C and D), but also because the molecular analysis of the serotypes A and C showed that the residues of C14R similarly interact with the polysaccharides of both types of capsules. In addition, it is important to consider that there are two additional motifs without predominance in a certain serotype plus glucuronoxylomanogalactan (GXMgal) [36], which may also interact with the peptide. While there are some studies on the use of AMPs against *C. neoformans* and *C. gattii*, these are less common than those reported for other yeasts such as *Candida* species. In addition, most available studies have generally tested only reference strains of one or both species. To our knowledge, this is the first time that an AMP was tested against isolates of all serotypes, recovered mostly from clinical samples and resistant to fluconazole (30% of the studied isolates), which strengthens our study [23,53,54].

Regarding the in vitro antifungal activity of C14R against the studied *Candida* species, it is worthy of notice that smaller concentrations of the peptide were needed to inhibit the growth of these ascomycete yeasts compared to the concentrations needed to inhibit cryptococcal species. One possible explanation could be the absence of capsule in *Candida* species, allowing for more direct entry of the peptide through the plasma membrane, in addition to the fact that C14R generates bigger size pores in *C. albicans* [25] compared to those formed in *C. neoformans*, perhaps increasing the availability of the peptide in the cytoplasm. Nevertheless, higher MIC values of a drug do not necessarily imply an inferior antifungal activity. For instance, when comparing the MIC values of susceptible isolates of *Cryptococcus* and *Candida* to amphotericin B and azoles, it has been established that, generally, the polyene is effective in lower concentrations against *Cryptococcus* species compared to *Candida* species, while the opposite occurs with both fluconazole and voriconazole [32].

The assessment of the antifungal activity of C14R against five additional species of *Candida* apart from *C. albicans* and *C. auris*, with the inclusion of several clinical isolates per species, further supports the potential use of the peptide to treat those species that are changing the global landscape of invasive candidiasis and candidemia. The use of C14R as a standalone to treat resistant isolates is also highlighted by the assessment of the peptide against species with intrinsic resistance to some antifungal drugs, such as *C. krusei*, and others with acquired resistance to fluconazole, voriconazole and caspofungin which were included in this study [9,10,11,12,55]. Moreover, while the early recognition of *Candida* infections is essential, considering that these are among the most common nosocomial bloodstream infections (BSI) in the world [56], an antimicrobial therapy is not always prompt and accurate. Therefore, the development of broad-spectrum antifungal agents or the implementation of combined regimens for the treatment of these mycoses should be of highest priority to improve patient outcomes and reduce the negative impact of these infections on human health. To date, several AMPs have shown promising in vitro results and have been positively tested in diverse animal models, predominantly in infections caused by *C. albicans*. In fact, several AMPs have gone through successful preclinical or clinical trials [23,24]. Other studies have shown that combinations of AMPs and currently used antifungal drugs potentiate the effect of both molecules, even for the treatment of intrinsically resistant species [24,25,57]. C14R, particularly, has been used in a hydrogel which serves as a wound dressing to help inhibit biofilm formation and eradicate *P. aeruginosa*, including a carbapenem-resistant strain [58]. Considering that *Candida* and in many cases *Cryptococcus* can form biofilms on medical devices, the same approach may be eventually used for the “capture-and-killing” of yeasts that aggravate the complications of unyielding infections in the human host and cause problems in surgical care.

## 5. Conclusions

Altogether, the potential pharmaceutical applications of C14R in fighting cryptococcosis and candidiasis are to be emphasized, although further investigations are still required. First, combinations of C14R and commonly used antifungal drugs could be tested, given that synergies between molecules are likely to improve the treatment outcome of these fungal infections, as has been reported elsewhere [25,57]. Second, it is necessary to consider that so far, only in vitro data is available, which does not necessarily reflect the conditions in vivo. Lastly, testing C14R against capsule-deficient mutants would clarify whether the capsule acts as a barrier or facilitator and whether the peptide disrupts capsule synthesis, which could substantially enhance the peptide’s therapeutic potential against cryptococcosis in vivo. Nevertheless, the pore-forming ability of C14R is a mechanism of action supporting both anti-cryptococcal and anti-candidal activity, which in addition makes the development of resistance difficult. Therefore, C14R can be considered for future therapeutic applications to fight the two fungal pathogens responsible for most cases of serious fungal diseases in the world.

## Figures and Tables

**Figure 1 pathogens-14-00511-f001:**
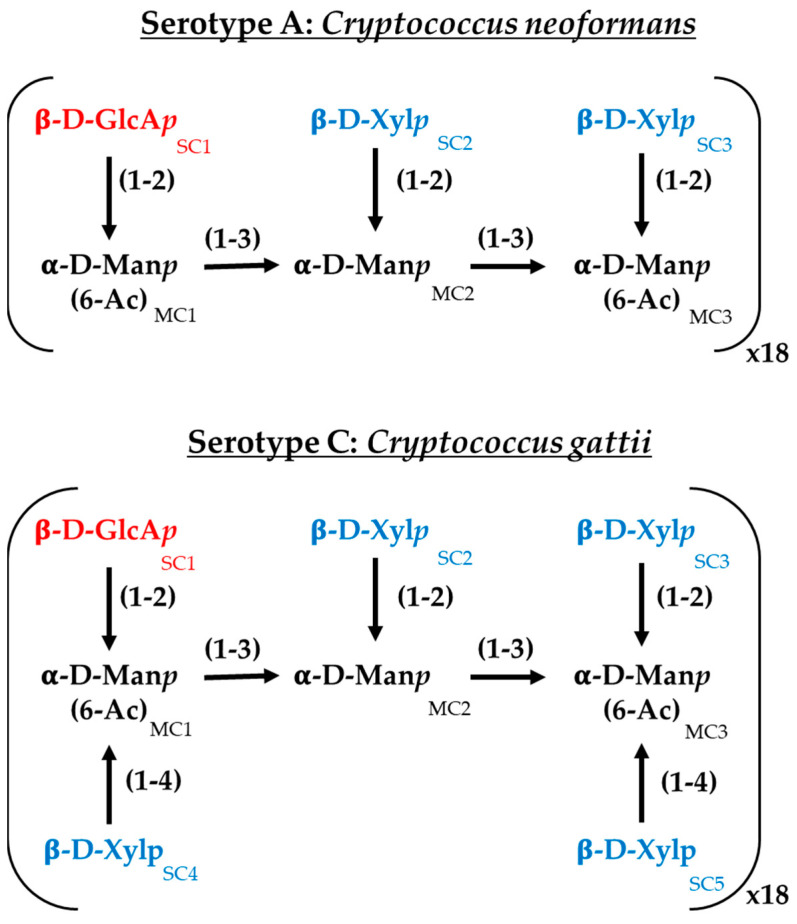
Chemical structure of the major polysaccharide glucuronoxylomannan (GXM) of the serotypes A (*Cryptococcus neoformans*) and C (*Cryptococcus gattii*) employed in this work. GlcA*p*: glucoronic acid, Xyl*p*: xylose; Man*p*: mannose.; MC: α-D-Man*p* residues and SC: sidechain residues.

**Figure 2 pathogens-14-00511-f002:**
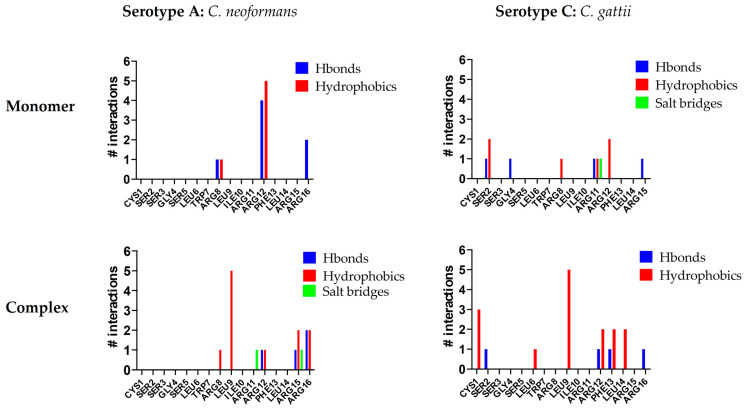
Interactions per residue of C14R with the polysaccharides of the capsules of the serotype A (*Cryptococcus neoformans*) and the serotype C (*Cryptococcus gattii*).

**Figure 3 pathogens-14-00511-f003:**
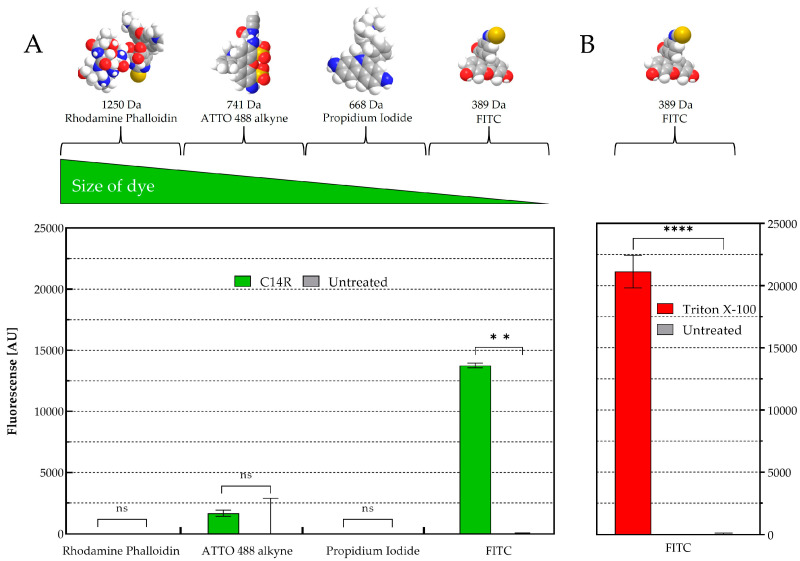
*Cryptococcus neoformans* membrane permeabilization ability of C14R. (**A**) Treatment with 25 µg/mL of C14R and staining of porous cells using rhodamine phalloidin (1250 Da), ATTO 488 alkyne (741 Da), propidium iodide (668 Da) or FITC (389 Da) as fluorescent dyes with untreated cells serving as negative controls and (**B**) using 0.4% Triton X-100 as positive control. The experiments were conducted in triplicate with error bars representing standard deviations. ns: not significant; ** statistically significant *p*-value < 0.01 and **** *p*-value < 0.0001.

**Table 1 pathogens-14-00511-t001:** Distribution, per species, of the minimum inhibitory concentration (MIC) values of C14R.

				No. of Isolates at MIC Value (μg/mL) ^1^							
Genus	Species	*n*	GM ^2^	0.391	0.781	1.563	3.125	6.25	12.5	25	50
*Cryptococcus*	*C. neoformans*	10	10.88					2	8		
	*C. gattii*	10	8.247				2	2	6		
*Candida*	*C. parapsilosis*	28	1.314	10	3	5	3	4	1		
	*C. glabrata*	18	1.503	8	2	2	1	1	3	-	1
	*C. tropicalis*	17	0.8476	8	2	4	3				
	*C. dubliniensis*	4	1.105	2	1	-	-	-	1		
	*C. krusei*	1	-	1							

^1^ The modal MIC for each distribution is underlined. ^2^ GM: Geometric mean in μg/mL calculated with four or more isolates.

**Table 2 pathogens-14-00511-t002:** Binding energy obtained with the score function of AutoDock Vina for the complexes C14R-cryptococcal capsules.

		AutoDock Vina Binding Energy (kcal/mol)	
Species	Serotype	Monomer	Complex
*C. neoformans*	A	−9.368	−11.525
*C. gattii*	C	−11.526	−11.682

## Data Availability

The original contributions presented in the study are included in the article, further inquiries can be directed to the corresponding author.

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
