# Peer review of "Broad Antifungal Spectrum of the Pore-Forming Peptide C14R Against Cryptococcus and Candida Species from the WHO Fungal Priority Pathogens List"

_pathogens, 2025, doi:10.3390/pathogens14060511_

Round 1
Reviewer 1 Report
Comments and Suggestions for Authors
Revision for the manuscript: Broad Antifungal Spectrum of the Pore-Forming Peptide C14R 2 against the Cryptococcus and Candida Species from the WHO 3 Fungal Priority Pathogens List
I would like to congratulate the authors on their work.
I just have a few questions and recommendations.
Candida glabrata, C. krusei and C. auris, as they are reclassified, should be mentioned in the article with their current name.
A table perhaps as supplementary material with all available information on the antifungal resistance of the isolates would be good. In lines 246-251 you mention that there is no correlation between the susceptibility of each species studied to C14R and fluconazole, but what about the echinocandins? The table could provide a better overview.
Since antimicrobial peptides are part of IS, what is the source of C14P, in which organism could we find this peptide?
Author Response
I would like to congratulate the authors on their work.
Answer. Thanks to the reviewer for the time to read and evaluate the manuscript and for the valuable comments and suggestions. In the revised manuscript and along with this point-by-point response, we aimed to address all issues mentioned by this and other reviewers.
I just have a few questions and recommendations.
Candida glabrata, C. krusei and C. auris, as they are reclassified, should be mentioned in the article with their current name.
Answer. Thanks to the reviewer for the suggestion. However, given that C. glabrata, C. krusei and C. auris were just recently taxonomically reallocated, as stated in the manuscript (lines 65-67), and that they are mentioned as such in the WHO priority list, we consider that they can still be referred to as “Candida” species. This is to facilitate the reading of the manuscript, from the title onwards, and to agree with the name of the disease (candidiasis), which has not been updated.
A table perhaps as supplementary material with all available information on the antifungal resistance of the isolates would be good. In lines 246-251 you mention that there is no correlation between the susceptibility of each species studied to C14R and fluconazole, but what about the echinocandins? The table could provide a better overview.
Answer. Thanks to the reviewer for the suggestion. We have included a supplementary table 1 with the distribution of all the MIC values of commonly used antifungals and C14R, against the studied isolates. There we highlighted resistant isolates. We have also added a sentence regarding the correlation between the susceptibility to C14R and echinocandins and other antifungals. Given that only a few isolates were resistant to other antifungals, no correlation between both molecules was found (tests not done).
Since antimicrobial peptides are part of IS, what is the source of C14P, in which organism could we find this peptide?
Answer. Thanks to the reviewer for the suggestion. In the introduction, we have added a sentence regarding the origin of C14R.
Reviewer 2 Report
Comments and Suggestions for Authors
In your title you can remove "the" before the cryptococcus and Candida spp.
- Your antifungal susceptibility testing. I miss the use of the gold standard treatment antifungals here. Please include the proper controls for the tested fungi, like ampB and fluconazole. This gives you a better comparison as your MIC values are relative, plus it gives your audience the ability to see of your compound has clinical potential, especially when compared to standard of care. The statement of "a good in vitro antifungal effect cannot be made if we do bot compare to a positive control. It is misleading.
- pore formation. Why was this only tested in cryptococcus? Also, where are your positive controls (ampB)? You need a proper positive control for again, reference, and also to show that the uptake of the dyes is due to pore formation and not just cell damage. Please show the histograms of this experiment also.
- Maybe you could give the title a thought. You only use Candida in your MIC assay and then proceed with crypto only, I think your title statement is too strong and the data does not support the candida part.
Please add the p-values in your graphs and tables. Please compare your results to a proper control.
Author Response
Answer. Thanks to the reviewer for the time to read and evaluate the manuscript and for the valuable comments and suggestions. In the revised manuscript and along with this point-by-point response, we aimed to address all issues mentioned by this and other reviewers.
1. In your title you can remove "the" before the cryptococcus and Candida spp.
Answer. The word “the” was removed from the tittle
2. Your antifungal susceptibility testing. I miss the use of the gold standard treatment antifungals here. Please include the proper controls for the tested fungi, like ampB and fluconazole. This gives you a better comparison as your MIC values are relative, plus it gives your audience the ability to see of your compound has clinical potential, especially when compared to standard of care. The statement of "a good in vitro antifungal effect cannot be made if we do bot compare to a positive control. It is misleading.
Answer. Thanks to the reviewer for the suggestion. The MIC values of commonly used antifungals against all studied isolates is now included in Table S1. In this table is possible to observe than even against resistant isolates, mainly to FCZ, the peptide has a good in vitro antifungal activity. The statement of “a good in vitro antifungal effect” is made considering that that geometric mean MIC of the peptide is low (compared to the MICs of other AMPs reported in the literature) and considering that none of the isolates was able to grow in the two highest concentrations of C14R tested (100 µg/ml and 200 µg/ml). In fact, the mode of C14R against cryptococcal species was 12.5 µg/ml, the sixth lowest concentration tested, and the mode of C14R against all Candida species was 0.390625 µg/ml, the lowest concentration tested. This indicates that the total growth of the bast majority of isolates of the studied species is inhibited with low concentrations of the peptide.
3. pore formation. Why was this only tested in cryptococcus? Also, where are your positive controls (ampB)? You need a proper positive control for again, reference, and also to show that the uptake of the dyes is due to pore formation and not just cell damage. Please show the histograms of this experiment also.
Answer. Thanks to the reviewer for the suggestion. As it was stated in the manuscript, pore formation was only tested in Cryptococcus given that this has been previously demonstrated by our group, in three Candida species. Cryptococcus was tested here because its unique characteristic of having a capsule (please refer to section 2.5). On the other hand, we agree with the reviewer, an appropriate positive control was missing and hence we performed additional experiments. However, amphotericin B would not be an appropriate control in this kind of assay. It has been previously described that MIC concentrations of this antifungal (2 µg/mL) kill completely all cells of C. neoformans, as such, in a flow cytometry-based assays, fluorescent dyes like propidium iodide are not able to penetrate into cryptococcal cells at reasonable amounts (please refer to Fernanda Sangalli-Leite et al. 2011; https://doi.org/10.1016/j.micinf.2011.01.015). Considering this, as positive control in our fluorescent-dye based permeabilization assay we now included Triton X-100 as a known pore-forming detergent. C. neoformans cells were treated for 2h with 0.4% Triton X-100 and the penetration of FITC was analyzed (we included only FITC, as this was the only dye being incorporated into C. neoformans after treatment with our peptide C14R) via fluorescence measurements using a microplate plate reader. In this newly added Figure 3B, FITC uptake was clearly observed after Triton treatment. For this type of experiment (measuring fluorescence in a microplate reader) no histograms can be made, which in contrast is for sure possible in flow cytometry.
4. Maybe you could give the title a thought. You only use Candida in your MIC assay and then proceed with crypto only, I think your title statement is too strong and the data does not support the candida part.
Answer. Thanks to the reviewer for this comment. However, we would like to clarify than antifungal susceptibility testing was done in both Candida and Cryptococcus species (please refer to section 2.3 and results section 3.1), showing that the total growth of the bast majority of isolates of the studied species (n=7, plus C. albicans and C. auris previously studied) is inhibited with low concentrations of the peptide, as such C14R has a broad antifungal spectrum. Also, pore formation has been demonstrated in three Candida species and one of Cryptococcus (in this study). In this way, we consider that the title can be kept as it is.
5. Please add the p-values in your graphs and tables. Please compare your results to a proper control.
Answer. Thanks to the reviewer for this comment. p-values are added in graphs and tables. Controls are also provided.
Reviewer 3 Report
Comments and Suggestions for Authors
The authors describe the molecular mode of action and the activity of an antimicrobial peptide on selected Candida and Cryptococcus species. The work is important and the findings are significant, as they demonstrate efficiency of the C14R peptide against a broader spectrum of opportunistic fungal pathogens. The introduction summarizes the current knowledge in the field well, and the experiments are carefully designed to provide a mechanistical explanation for the observed antifungal activity. Methods are described well, and data analysis/presentation are appropriate. The discussion summarizes the findings appropriately and the conclusions make sense. The manuscript has been carefully written and edited, and beside 2 typographical errors (capitalization of SER in in line 287; “that” instead of “than” in line 377), I find nothing to complain.
Author Response
The authors describe the molecular mode of action and the activity of an antimicrobial peptide on selected Candida and Cryptococcus species. The work is important and the findings are significant, as they demonstrate efficiency of the C14R peptide against a broader spectrum of opportunistic fungal pathogens. The introduction summarizes the current knowledge in the field well, and the experiments are carefully designed to provide a mechanistical explanation for the observed antifungal activity. Methods are described well, and data analysis/presentation are appropriate. The discussion summarizes the findings appropriately and the conclusions make sense. The manuscript has been carefully written and edited, and beside 2 typographical errors (capitalization of SER in in line 287; “that” instead of “than” in line 377), I find nothing to complain.
Answer. Thanks to the reviewer for the time to read and evaluate the manuscript and for the valuable comments and suggestions. In the revised manuscript and along with a point-by-point response, we aimed to address all issues mentioned by this and other reviewers.
Both typographical errors were corrected.
Round 2
Reviewer 2 Report
Comments and Suggestions for Authors
Thanks for the clarification. Congrats on your work!